# N6-Methyladenosine Positively Regulates Coxsackievirus B3 Replication

**DOI:** 10.3390/v16091448

**Published:** 2024-09-11

**Authors:** Hainian Zhao, Zhiyun Gao, Jiawen Sun, Hongxiu Qiao, Yan Zhao, Yan Cui, Baoxin Zhao, Weijie Wang, Sandra Chiu, Xia Chuai

**Affiliations:** 1Department of Pathogenic Biology, Hebei Medical University, Shijiazhuang 050017, China; 23031100026@stu.hebmu.edu.cn (H.Z.); 17800577@hebmu.edu.cn (Z.G.); sunjiawenwy@163.com (J.S.); 18200787@hebmu.edu.cn (Y.Z.); zhaobx@wh.iov.cn (B.Z.); weijiewang@hebmu.edu.cn (W.W.); 2Experimental Center for Teaching, Hebei Medical University, Shijiazhuang 050017, China; 3State Key Laboratory of Virology, Wuhan Institute of Virology, Center for Biosafety Mega Science, Chinese Academy of Sciences, Wuhan 430207, China; 4Division of Life Sciences and Medicine, University of Science and Technology of China, Hefei 230026, China

**Keywords:** coxsackievirus B3, N6-methyladenosine, replication, m^6^A-related proteins

## Abstract

Enteroviruses such as coxsackievirus B3 are identified as a common cause of viral myocarditis, but the potential mechanism of its replication and pathogenesis are largely unknown. The genomes of a variety of viruses contain N6-methyladenosine (m^6^A), which plays important roles in virus replication. Here, by using the online bioinformatics tools SRAMP and indirect immunofluorescence assay (IFA), we predict that the CVB3 genome contains m^6^A sites and found that CVB3 infection could alter the expression and cellular localization of m^6^A-related proteins. Moreover, we found that 3-deazaadenosine (3-DAA), an m^6^A modification inhibitor, significantly decreased CVB3 replication. We also observed that the m^6^A methyltransferases methyltransferase-like protein 3 (METTL3) and METTL14 play positive roles in CVB3 replication, whereas m^6^A demethylases fat mass and obesity-associated protein (FTO) or AlkB homolog 5 (ALKBH5) have opposite effects. Knockdown of the m^6^A binding proteins YTH domain family protein 1 (YTHDF1), YTHDF2 and YTHDF3 strikingly decreased CVB3 replication. Finally, the m^6^A site mutation in the CVB3 genome decreased the replication of CVB3 compared with that in the CVB3 wild-type (WT) strain. Taken together, our results demonstrated that CVB3 could exploit m^6^A modification to promote viral replication, which provides new insights into the mechanism of the interaction between CVB3 and the host.

## 1. Introduction

Coxsackievirus B3 (CVB3) belongs to the *Enterovirus* genus of *Picornaviridae* and has been long studied in models of viral myocarditis [1]. Acute VMC may cause sudden cardiac death or progress to dilated cardiomyopathy (DCM) or even heart failure [2]. Currently, no effective treatments or vaccines are available, and the mechanisms of CVB3 infection are not fully elucidated. Uncovering the potential mechanisms of CVB3-induced myocardial injury is crucial for the development of therapeutic strategies. CVB3 is a nonenveloped virus and possesses a 7.4 kb single positive-stranded RNA. The CVB3 genome encodes four structural proteins (VP1, VP2, VP3 and VP4) and seven nonstructural proteins (2A, 2B, 2C, 3A, 3B, 3C and 3D). VP1, VP2, VP3 and VP4 form the viral capsid structure and are responsible for binding to the receptor of the host cell. Among the seven nonstructural proteins, proteases 2A and 3C are responsible for the processing of viral polyproteins, RNA-dependent RNA polymerase 3D is involved in viral genome replication and 2B and 2C participate in RNA synthesis [1,3].

N6-methyladenosine (m^6^A) is the most abundant internal modification of eukaryotic mRNAs among a variety of known RNA modifications, and it regulates many biological processes of RNA [4], including splicing [5], nuclear export [6], stability [7,8] and translation efficiency [9]. m^6^A modifications of mRNAs are mostly located at translation start sites, stop codons, and 3′ untranslated regions (3′ UTRs) [10,11,12] and occur at the consensus motif [G/A/U][G>A]m^6^AC[U>A>C] [13,14,15]. m^6^A modification is a dynamically reversible process. m^6^A is added to mRNA by a methyltransferase complex containing METTL3 and METTL14 and the cofactor Wilms’ tumour 1-associated protein (WTAP) [12,16,17], and is removed from mRNA by the demethylases FTO or ALKBH5 [18,19]. m^6^A-binding proteins such as the YTH domain-containing protein YTHDF1-3 recognize and bind to this modification and importantly affect the functions of m^6^A [12].

Several groups have shown that internal m^6^A modifications are present in viral RNA and play proviral or antiviral roles during the replication cycle of viruses [20,21]. For example, m^6^A modification positively regulates enterovirus type 71 (EV71) replication [22]. During EV71 replication, METTL3 interacts with the viral RNA-dependent RNA polymerase 3D and induces increased SUMOylation and ubiquitination of the 3D that boosted viral replication, as 3D is ubiquitinated in a SUMOylation-dependent manner, such that the interplay of SUMOylation and ubiquitination enhances the stability of the viral polymerase [23]. However, m^6^A dynamics negatively regulate the production or release of infectious hepatitis C virus (HCV) viral particles [24]. During HCV infection, YTHDF proteins relocate to lipid droplets, sites where viral RNA synthesis and viral particle assembly occur, to interfere with viral RNA packaging and the production of infectious viral particles. Taken together, these studies demonstrate that m^6^A modification plays important roles in virus replication.

m^6^A modification has not yet been identified in CVB3 RNA, and the mechanism by which m^6^A regulates CVB3 replication remains unclear. Since EV71 is also a member of the Enterovirus genus, which shares a similar structure and genome with CVB3, we speculate that m^6^A modification could also regulate CVB3 replication. Here, we explored the regulatory effect of m^6^A modification on CVB3 infection in HeLa cells and reported that it positively regulates CVB3 replication by altering the m^6^A level or the expression of m^6^A-related proteins. This study provides novel insights into CVB3 infection and pathogenesis, which may help in the development of effective strategies against CVB3.

## 2. Materials and Methods

### 2.1. Cells and Viruses

HeLa cells were grown in Dulbecco’s modified Eagle’s medium (DMEM; Gibco) supplemented with 10% FBS. The cells were grown in a humidified atmosphere with 5% CO_2_ at 37 °C. The plasmid pMKS1-CVB3-eGFP, containing the full-length genome of the CVB3 pH 3 strain (GenBank U57056.1) [25] and a green fluorescence protein (GFP) gene [26], was kindly provided by Dr. Ralph Feuer of the Scripps Research Institute (La Jolla, CA, USA). Plasmid DNA was transfected into HEK293T cells by using Lipofectamine^®^ 2000 reagent (Thermo Fisher, Waltham, MA, USA) according to the manufacturer’s protocol. CVB3 viruses were amplified and titrated by plaque-forming units (PFUs) in HeLa cells.

### 2.2. Western Blot

The cells were lysed in RIPA buffer (Bestbio, Shanghai, China) supplemented with protease inhibitor cocktails (Bestbio), and 20 µg of total protein was subjected to 10% SDS polyacrylamide gel electrophoresis. The samples were transferred to PVDF Western Blotting membranes (Thermo Fisher) and blotted with anti-VP1 (Dako-M7064, Glostrup, Denmark), anti-METTL3 (15073-1-AP, Proteintech, Wuhan, China), anti-METTL14 (26158-1-AP, Proteintech), anti-FTO (27226-1-AP, Proteintech), anti-ALKBH5 (16837-1-AP, Proteintech), anti-YTHDF1 (17479-1-AP, Proteintech), anti-YTHDF2 (24744-1-AP, Proteintech), anti-YTHDF3 (25537-1-AP, Proteintech) and β-tubulin (66240-1-Ig, Proteintech) antibodies. The corresponding horseradish peroxidase (HRP)-conjugated AffiniPure goat anti-mouse IgG (H+L) (SA00001-1, Proteintech) and the corresponding horseradish peroxidase (HRP)-conjugated AffiniPure goat anti-rabbit IgG (H+L) (SA00001-2, Proteintech) were used as secondary antibodies. The membranes were analysed with the WesternLumaxLightTM Superior (ZETA) using an enhanced chemiluminescence (ECL) imager (Gene).

### 2.3. Indirect Immunofluorescence Assay (IFA)

HeLa cells were seeded in six-well plates 1 day before infection at ∼50% confluence, after which the cells were infected with CVB3 (MOI = 1) and incubated for 24 h. Then, the cells were washed three times with PBS, fixed in 4.0% paraformaldehyde for 15 min, permeabilized in 0.5% Triton X-100 for approximately 20 min and blocked in 10% goat serum for 1 h at room temperature. The cells were incubated with primary antibodies at the dilutions suggested by the manufacturers overnight at 4 °C, washed three times with PBS and stained with the appropriate secondary antibodies for 1 h at room temperature. Nuclei were stained with Hoechst. The slides were observed under an Olympus confocal microscope.

### 2.4. Quantitative Reverse-Transcription PCR (qRT–PCR)

Total RNA was extracted via TRIzol reagent (TIANGEN, Beijing, China). Reverse transcription was performed with 1 µg of total RNA via HiScript III reverse transcriptase (Vazyme, Nanjing, China). qRT–PCR was performed via chamQ universal SYBR qPCR Master Mix (Vazyme) on an Applied BiosystemsTM QuantStudioTM 3 (Thermo Fisher). Relative gene expression levels were obtained by normalizing the cycle threshold (Ct) values to those of HS-ACTB to yield 2−△△Ct values. The primers used for gene expression were as follows: CVB3 (forward: 5′-CGA TCA ACA GTC AGC GTG G-3′, reverse: 5′-TGG CCG GAT AAC GAA CGC-3′) and HS-ACTB (forward: 5′-CCT GGC ACC CAG CAC AAT-3′, reverse: 5′-GGG CCG GAC TCG TCA TAC-3′). Amplification was carried out at 95 °C for 3 min; 35 cycles of 95 °C for 15 s, 55 °C for 30 s, and 72 °C for 1 min; with a final extension of 72 °C for 10 min.

### 2.5. Generation of Stable Cell Lines

The expression of METTL3, METTL14, FTO, ALKBH5, YTHDF1, YTHDF2 and YTHDF3 was knocked down by short hairpin RNA (shRNA) targeting a human-specific gene or a nonspecific oligonucleotide ligated into the lentiviral expression vector pCDH. shRNAs specific to each gene used in the study are as follows: shMETTL3: 5′-GCAAGAATTCTGTGACTATGG-3′, shMETTL14: 5′-GCTTAACCCATTAGTACTATC′, shFTO: 5′-GCTGAAATATCCTAAACTAAT-3′, shALKBH5: 5′-GTCCGTGTCCTTCTTTAGCGA-3′, shYTHDF1: 5′-GGATACAGTTCATGACAATGA-3′, shYTHDF2: 5′-GGTTCTGGATCTACTCCTTCA-3′,shYTHDF3: 5′-GCAATGATACTTTGAGTAAGG-3′. shNC: 5′-GGTGAAGGTGATGCAACATAC-3′. Lentiviruses were produced via the cotransfection of pCDH and the packaging plasmids pLP1, pLP2, and pVSV-G into 293T cells. Pseudoviral particles were subsequently used to infect HepG2 cells. To obtain stably transfected cell lines, the cells were treated with puromycin. The knockdown efficiency was confirmed by qPCR and Western blotting, and the cells were used for subsequent experiments.

### 2.6. MeRIP-qPCR

The MeRIP assay was performed using a m^6^A RNA enrichment kit (Epigentek, NY, USA). Briefly, the target fragment containing m^6^A regions was pulled down via a bead-bound m^6^A capture antibody, and the enriched RNA was then released, purified and eluted. Finally, qRT–PCR was performed to quantify the changes in the methylation of target genes.

### 2.7. Cell Counting Kit-8 (CCK-8)

HeLa cells were seeded in 96-well plates at 5000 cells/well in DMEM supplemented with 10% FBS. Approximately 18 h later, the media were replaced with 100 µL/well media containing various concentrations of 3-DAA (HY-W013332, MCE). After 48 h of cell culture at 3-DAA, the medium was carefully removed, 100 µL of culture medium containing CCK8 (HY-K0301-3000T, MCE) solution in DMEM supplemented with 10% FBS was added to each well, and the cells were further incubated for an additional 4 h at 37 °C. The absorbance at 450 nm was read with a BioTeK microplate reader (Agilent Technologies. Santa Clara, CA, USA).

### 2.8. Statistical Analysis

The data were analysed via unpaired Student’s *t* test with Prism software (v10.2.0), and statistical significance was defined as *p* < 0.05.

## 3. Results

### 3.1. m^6^A Modification Regulates CVB3 Replication

To investigate whether CVB3 RNA was m^6^A modified, we first utilized the online bioinformatics tool SRAMP (http://www.cuilab.cn/sramp/, accessed on 16 May 2023) to predict m^6^A sites and found that the CVB3 RNA genome contained 28 m^6^A modification sites, including one very high confidence site, seven high confidence sites, nine moderate confidence sites, and eleven low confidence sites (Figure 1A and Appendix A).

It is universally acknowledged that 3-deazaadenosine (3-DAA) blocks m^6^A modification of mRNAs by decreasing S-adenosyl-L-methionine (SAM) [27,28,29], which may influence the replication of viruses, such as influenza A virus (IAV) [30], herpes virus type 1 (HSV-1) [31] and Kaposi’s sarcoma-associated herpesvirus (KSHV) [32]. Next, we investigated the effect of 3-DAA on CVB3 replication. We first detected the viability of HeLa cells treated with various concentrations of 3-DAA for 48 h. We found that under 20 µM and 40 µM 3-DAA, the cell survival rates were 97.02% and 86.85%, respectively, which were not significantly different from those of the control group (Appendix A). Subsequently, 20 µM and 40 µM 3-DAA were chosen for the following experiments. CVB3 RNA copies, the VP1 protein and the viral titre were measured via reverse transcription and quantitative PCR (RT–qPCR), Western blotting and plaque-forming unit (PFU) assays, respectively. In addition, CVB3 GFP expression was monitored via fluorescence microscopy, and the fluorescence intensity was quantified via ImageJ software (version 1.53c). The results revealed a significant decrease in the number of CVB3 RNA copies, VP1 protein expression, viral titre and fluorescence intensity following treatment with 3-DAA (Figure 1B–E). Taken together, these findings suggest m^6^A modifications may affect CVB3 replication.

### 3.2. CVB3 Infection Modulates the Expression Patterns of m^6^A-Related Proteins

To further verify the relationship between m^6^A modification and CVB3 replication, we investigated the expression and cellular localization of host m^6^A-related proteins following CVB3 infection. Western blotting and immunofluorescence staining were utilized for these analyses. We noted a slight decrease in METTL3 and METTL14 protein levels at 36 h post-infection (hpi), whereas the protein levels of FTO and ALKBH5 did not significantly change during CVB3 infection. Additionally, the protein levels of YTHDF1, YTHDF2 and YTHDF3 significantly decreased at 36 hpi (Figure 2A). For the cellular localization of host m^6^A-related proteins, the immunofluorescence results revealed that METTL3, METTL14, FTO, ALKBH5 and YTHDF3 translocated from the nucleus to the cytoplasm following CVB3 infection (Figure 2B–E,H). Conversely, both YTHDF1 and YTHDF2 were predominantly located in the cytoplasm in both uninfected and infected HeLa cells (Figure 2F,G). These findings suggested that CVB3 infection modulated the protein expression and subcellular distribution of m^6^A methyltransferases, demethylases and binding proteins.

### 3.3. METTL3 and METTL14 Enhance CVB3 Replication in HeLa Cells

To evaluate the effect of m^6^A modification on CVB3 replication, we first analysed the role of the methyltransferase complex in the replication of CVB3 by knocking down METTL3 and METTL14 in HeLa cells, followed by CVB3 infection. We knocked down endogenous METTL3 and METTL14 in HeLa cells via shRNA, and the expression of both proteins was assessed via western blot (Appendix A). We found that, compared with the control, the knockdown of METTL3 and METTL14 resulted in substantial decreases in CVB3 RNA copy numbers, VP1 protein synthesis, viral titres and GFP expression (Figure 3A–H). Taken together, our results suggest that the m^6^A methyltransferases METTL3 and METTL14 favour CVB3 replication by enhancing the expression of CVB3 RNA, VP1 protein synthesis and the production of infectious particles.

### 3.4. FTO and ALKBH5 Inhibit CVB3 Replication in HeLa Cells

We next examined the role of m^6^A erasers in the replication of CVB3 by knocking down endogenous FTO and ALKBH5 in HeLa cells, followed by CVB3 infection. We knocked down endogenous FTO and ALKBH5 in HeLa cells via shRNA (Appendix A). In contrast to the results of the knockdown of METTL3 and METTL14, the knockdown of FTO and ALKBH5 resulted in significant increases in the levels of CVB3 RNA, VP1 protein and viral titres (Figure 4A–C,E–G). Nevertheless, we observed nonsignificant changes in GFP expression (Figure 4D,H). Collectively, these results suggest that the m^6^A demethylases FTO and ALKBH5 negatively regulate CVB3 replication.

### 3.5. YTHDF1, YTHDF2 and YTHDF3 Interference with the Expression of CVB3 RNA

Owing to the important functions of m^6^A-binding proteins in recognizing m^6^A modifications [12], we further investigated the functions of the m^6^A readers YTHDF1, YTHDF2 and YTHDF3 in CVB3 replication. The knockdown efficiency of YTHDF1, YTHDF2 and YTHDF3 in HeLa cells was confirmed by Western blotting (Appendix A). Upon infection with CVB3, the results revealed that the knockdown of YTHDF1 and YTHDF2 resulted in significant reductions in CVB3 RNA copies, VP1 proteins, viral titres and GFP expression, whereas the knockdown of YTHDF3 led to a decrease in only CVB3 RNA copies and GFP expression (Figure 5A–L). These results collectively suggested that YTHDF1, YTHDF2 and YTHDF3 could interfere with the expression of CVB3 RNA during CVB3 infection.

### 3.6. Defects in the Replication of m^6^A-Abrogating CVB3 Mutants

Finally, we abrogated a certain m^6^A site in the CVB3 genome to verify the influence of m^6^A modification. We chose a very high confidence site in the SRAMP prediction results. We subsequently mutated the C residue at nt 4519 without changing the amino acid (Figure 6A). m^6^A-modified RNA immunoprecipitation (MeRIP–qPCR) revealed a robust decrease in the relative m^6^A level of the 4519 m^6^A mutant compared with that of WT CVB3 (Figure 6B). Notably, m^6^A mutants presented lower viral titres than did CVB3 WT. Therefore, m^6^A modification of the CVB3 genome may positively regulate CVB3 replication (Figure 6C).

## 4. Discussion

m^6^A modification is an important post-transcriptional regulatory mechanism that plays important roles in the replication of several viruses. Here, we predict that the CVB3 genome contains m^6^A modifications via SRAMP analysis and that the m^6^A modification inhibitor 3-DAA could significantly inhibit CVB3 replication. In addition, our results suggested that METTL3 and METTL14 enhanced CVB3 replication during CVB3 infection; however, FTO and ALKBH5 had opposite effects. We also found that the m^6^A reader proteins YTHDF1, YTHDF2 and YTHDF3 promoted CVB3 replication by affecting the expression of CVB3 RNA. In the end, we observed that the m^6^A mutants of CVB3 inhibited the replication of CVB3 after mutation of the m^6^A site. In summary, our results suggest that m^6^A modification plays an important role in CVB3 replication.

m^6^A has been found in the viral genome for decades, but the regulatory function of m^6^A modification has only been recognized in recent years [33,34]. m^6^A modification affects viral replication and strongly affects infection outcomes. Several studies have reported that m^6^A modifications, such as those of HIV [35], IAV [36], HCV [24], ZIKV [37] and EV71 [22], are present in the viral genome. Accordingly, we utilized the online bioinformatics tool SRAMP to investigate whether CVB3 RNA was m^6^A modified. The results revealed that the CVB3 genome also has m^6^A modifications, which are similar to those of other viruses. Further studies need to use MeRIP-seq to confirm the m^6^A sites in the CVB3 genome.

Relevant studies have shown that virus infection alters the expression patterns of m^6^A-related proteins [22,38]. We observed that the protein levels of METTL3 and METTL14 decreased slightly and that the protein levels of YTHDF1, YTHDF2 and YTHDF3 clearly decreased compared with those in uninfected HeLa cells. METTL3, METTL14, FTO, ALKBH5 and YTHDF3 spread from the nucleus to the cytoplasm after CVB3 infection. However, the subcellular localization of YTHDF1 and YTHDF2 was not altered after CVB3 infection. These results illustrated that CVB3 infection altered the protein expression and subcellular localization of m^6^A-related proteins, which is worth investigating as the potential mechanism involved in CVB3 replication. We detected the colocalization of CVB3 VP1 and m^6^A-related proteins. As an RNA virus, CVB3 replicates in the cytoplasm, and the transition of m^6^A-related proteins suggests that CVB3 RNA may undergo m^6^A methylation and that the replication of CVB3 may be regulated by m^6^A modification.

m^6^A modification plays a proviral or antiviral role during virus infection and can positively regulate IAV [30], EV71 [27] and HMPV [39] replication; negatively regulate HCV [24] and ZIKV [37] replication; and positively and negatively regulate the replication of HIV [35,40,41,42] and HRSV [42,43]. In this study, we found that inhibition of the m^6^A level by 3-DAA significantly decreased the RNA copy number, the expression of the VP1 protein and the viral titre of CVB3. We observed that knockdown of the m^6^A writer proteins METTL3 and METTL14 had the same effects, while knockdown of the m^6^A erasers FTO and ALKBH5 promoted CVB3 replication by affecting CVB3 mRNA synthesis, protein synthesis and viral release. These results suggest that m^6^A modification may play a proviral role in CVB3 infection. In addition, we observed that YTHDF1, YTHDF2 and YTHDF3 regulated CVB3 replication by affecting CVB3 mRNA synthesis. These results are consistent with recent reports that the m^6^A machinery positively regulates IAV, EV71 and HMPV. Moreover, the influence of other m^6^A-related proteins requires further investigation.

Furthermore, after the m^6^A site in the CVB3 genome was mutated, we observed that m^6^A modification in the CVB3 genome led to a robust reduction in viral titre and relative m^6^A level compared with those in the WT genome. These results were similar to those for EV71 and IAV. These results also suggest that m^6^A modification of nt 4518 in the CVB3 genome affects the release of virus particles. Accordingly, m^6^A modification in the CVB3 genome plays an important role in CVB3 replication.

## 5. Conclusions

In conclusion, our results suggest that m^6^A modification positively regulates CVB3 replication. m^6^A-related proteins play different roles during CVB3 replication by affecting CVB3 mRNA synthesis, protein synthesis and viral release. Nevertheless, the underlying regulatory mechanisms remain needing to be further explored. The processes related to CVB3 replication, for example, could be affected by m^6^A modification of the CVB3 genome. The current study provides an experimental basis and therapeutic targets for the treatment of infectious diseases caused by CVB3 infection.

## Figures and Tables

**Figure 1 viruses-16-01448-f001:**
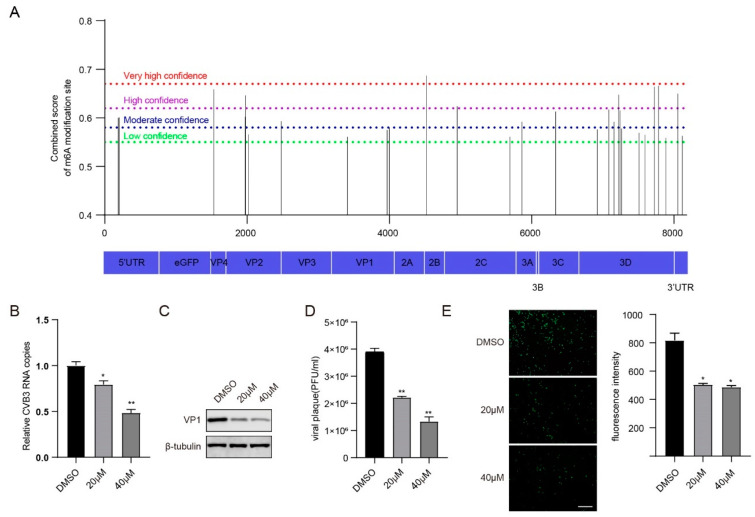
m^6^A modification regulates CVB3 replication. (**A**) Prediction of m^6^A modification sites on CVB3 genomic RNA via the online bioinformatics tool SRAMP. (**B**) Total RNA was extracted at the indicated times from CVB3-infected HeLa cells, in which HeLa cells were pretreated with the indicated concentrations of 3-DAA for 24 h. CVB3 RNA was quantified via qRT–PCR, with HS-ACTB used as a control. * *p* < 0.05, ** *p* < 0.01. (**C**) HeLa cells were pretreated with the indicated concentrations of 3-DAA for 24 h before infection with CVB3. Western blotting was carried out to determine the expression of VP1. (**D**) HeLa cells were pretreated with the indicated concentrations of 3-DAA for 24 h before infection with CVB3, and the supernatants were collected at the indicated times post-infection to measure virus titres as plaque-forming units (PFUs). ** *p* < 0.01. (**E**) Fluorescence microscopy images of CVB3-infected HeLa cells. GFP expression was monitored at the indicated times via fluorescence microscopy. Micrographs at ×10 magnification (scale bar of 100 μm) are shown. The fluorescence intensity was quantified with ImageJ software. * *p* < 0.05.

**Figure 2 viruses-16-01448-f002:**
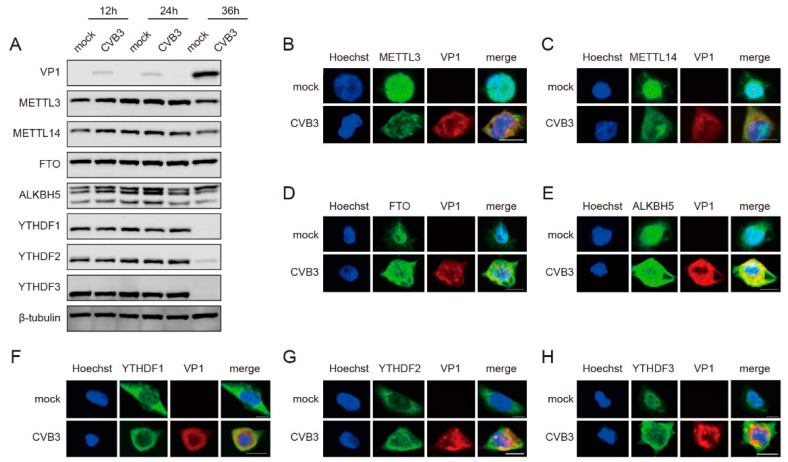
CVB3 infection alters the expression patterns of m^6^A-related proteins. (**A**) HeLa cells infected with CVB3 (MOI = 0.01) were harvested at 12 hpi, 24 hpi and 36 hpi. The expression of m^6^A-related proteins was detected by Western blotting. (**B**–**H**) Confocal microscopy images of CVB3- or mock-infected HeLa cells at 24 h post-infection (hpi). The nucleus (blue) and virus protein (red) were labelled with Hoechst and a VP1-specific antibody, respectively. The m^6^A-related proteins (green) were stained with antibodies as indicated. Scale bars, 5 μm.

**Figure 3 viruses-16-01448-f003:**
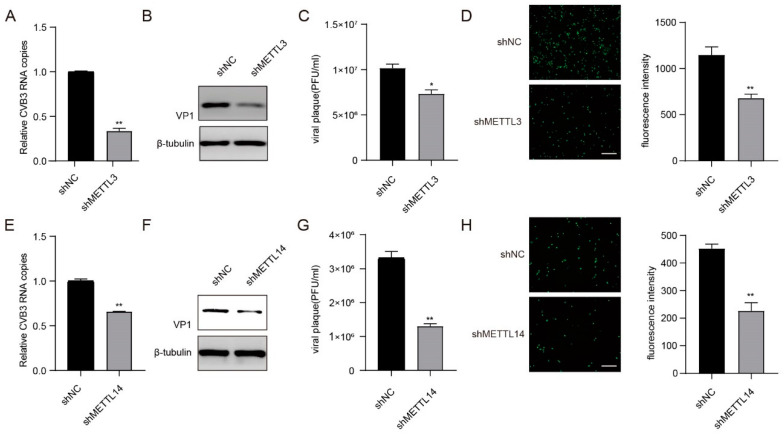
METTL3 and METTL14 enhance CVB3 replication in HeLa Cells. (**A**,**E**) Total RNA was extracted from CVB3-infected HeLa cells in which METTL3 or METTL14 was knocked down at the indicated times. CVB3 RNA was quantified via qRT–PCR, with HS-ACTB used as a control. ** *p* < 0.01. (**B**,**F**) VP1 protein expression when METTL3 and METTL14 were knocked down in HeLa cells. (**C**,**G**) HeLa cells in which METTL3 or METTL14 was knocked down were infected with CVB3, and the supernatants were collected at the indicated times post-infection to measure virus titres as PFU. * *p* < 0.05, ** *p* < 0.01. (**D**,**H**) Fluorescence microscopy images of CVB3-infected HeLa cells. GFP expression was monitored at the indicated times via fluorescence microscopy. Micrographs at ×10 magnification (scale bar of 100 μm) are shown. The fluorescence intensity was quantified with ImageJ software. ** *p* < 0.01.

**Figure 4 viruses-16-01448-f004:**
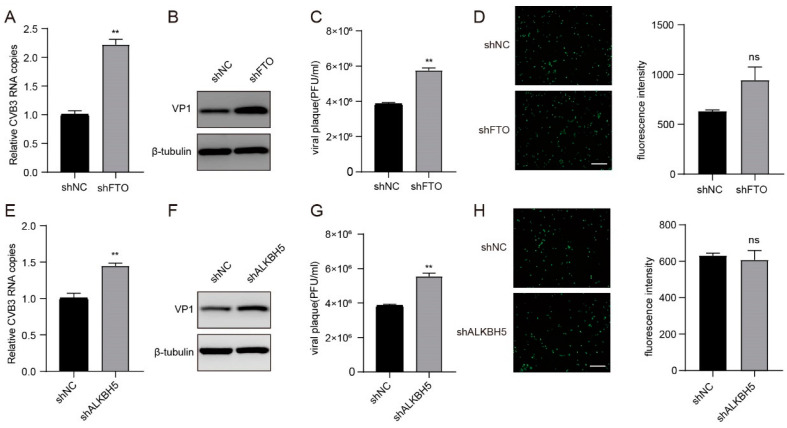
FTO and ALKBH5 inhibit CVB3 replication in HeLa cells. (**A**,**E**) Total RNA was extracted from CVB3-infected HeLa cells in which FTO or ALKBH5 was knocked down at the indicated times. CVB3 RNA was quantified via qRT–PCR, with HS-ACTB used as a control. ** *p* < 0.01. (**B**,**F**) FTO and ALKBH5 were knocked down in HeLa cells, followed by CVB3 infection, and the expression of VP1 was assessed by Western blotting. (**C**,**G**) HeLa cells in which FTO or ALKBH5 was knocked down or not knocked down were infected with CVB3, and the supernatants were collected at the indicated times post-infection to measure virus titres as PFU. ** *p* < 0.01. (**D**,**H**) Fluorescence microscopy images of CVB3-infected HeLa cells. GFP expression was monitored at the indicated times via fluorescence microscopy. Micrographs at ×10 magnification (scale bar of 100 μm) are shown. The fluorescence intensity was quantified with ImageJ software.

**Figure 5 viruses-16-01448-f005:**
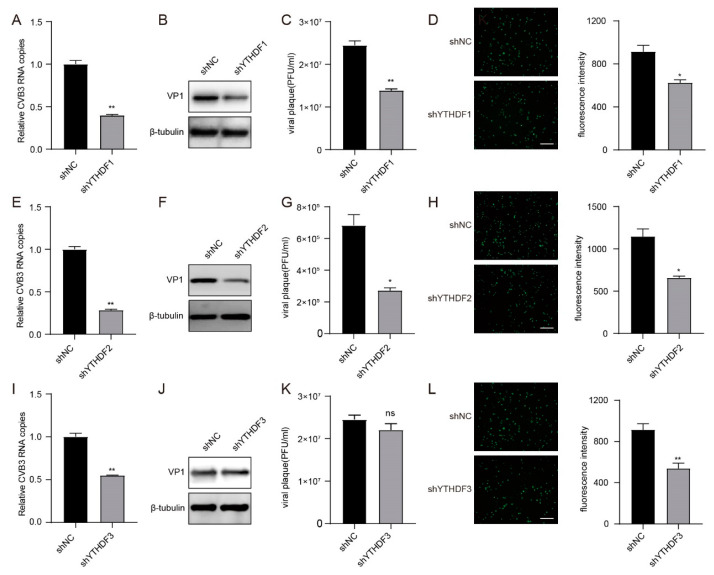
YTHDF1, YTHDF2 and YTHDF3 interfere with the expression of CVB3 RNA. (**A**,**E**,**I**) Total RNA was extracted at the indicated times from CVB3-infected HeLa cells in which YTHDF1, YTHDF2 or YTHDF3 was knocked down. CVB3 RNA was quantified via qRT–PCR, with HS-ACTB used as a control. ** *p* < 0.01. (**B**,**F**,**J**) YTHDF1, YTHDF2 and YTHDF3 were knocked down in HeLa cells, and Western blotting was carried out to determine the expression of VP1 after CVB3 infection. (**C**,**G**,**K**) HeLa cells in which YTHDF1, YTHDF2 or YTHDF3 was knocked down were infected with CVB3, and the supernatants were collected at the indicated times post-infection to measure virus titres as PFU. * *p* < 0.05, ** *p* < 0.01. (**D**,**H**,**L**) Fluorescence microscopy images of CVB3-infected HeLa cells. GFP expression was monitored at the indicated times via fluorescence microscopy. Micrographs at ×10 magnification (scale bar of 100 μm) are shown. The fluorescence intensity was quantified with ImageJ software. * *p* < 0.05, ** *p* < 0.01.

**Figure 6 viruses-16-01448-f006:**
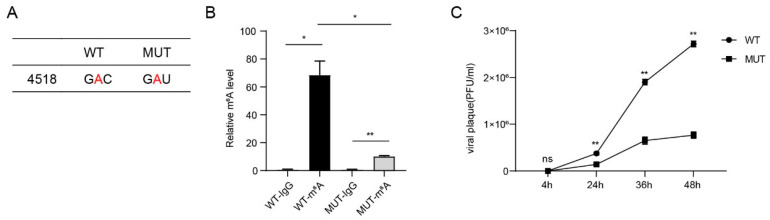
m^6^A-abrogating CVB3 mutants have defects in replication. (**A**) Diagram of wild-type (WT) and m^6^A mutant CVB3. The sequences of the WT and mutant strains are presented. (**B**) MeRIP-qPCR was used to assess the RNA m^6^A modification of wild-type (WT) and m^6^A mutant CVB3. The enrichment of m^6^A in each group was calculated by m^6^A IP/input and IgG IP/input. * *p* < 0.05, ** *p* < 0.01. (**C**) Viral titres (PFU/mL) at 4, 24, 36 and 48 hpi. HeLa cells were infected with CVB3 wild-type (WT) or m^6^A mutant strains, and the supernatants were collected at the indicated times post-infection to measure virus titres as PFU. * *p* < 0.05, ** *p* < 0.01.

## Data Availability

All the relevant data are provided in this paper.

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
