# Peer review of "N6-Methyladenosine Positively Regulates Coxsackievirus B3 Replication"

_viruses, 2024, doi:10.3390/v16091448_

Round 1

Reviewer 1 Report

Comments and Suggestions for Authors

This study by Zhao et al explores the involvement of N6-methyladenosine in coxsackievirus B3 replication. This is a something that has been interrogated in many other viruses, including enterovirus-71 (a very similar virus to CVB3), thus the impact is modest but still interesting. Generally, the manuscript is well-written, experimental approach is sound, and the data mostly support the conclusions that m6A is important for CVB3 replication. However, the rigor of the data in certain spots needs to be improved. 

Major comments:

Western blots are not quantified and only show a single lane for each group. These should be done in replicate and quantified to better support protein data. If the images are representative of the quants, then only showing single bands is fine.

Figure 2A: At 36h PI, this MOI of CVB3 infection generally kills all the cells in culture. Thus it is difficult to ascertain if drops in protein levels are due to virally-mediated pathways so-to-speak, or if only cell corpses are being analyzed. Also, if YTHDF proteins are lower at this point, it would be interesting to see if this is transcriptional inhibition or if this may be due to proteolysis. It is also unclear at what point post-infection the other panels represent (and the rest of the figures). Additionally, single cell images are not much to go off of without any type of quantification assessing a larger population of cells.

Figure 4D and H: It is interesting that viral GFP levels are not significantly changed. It would be good to provide some thoughts on what is going on there.

Figure 5J: Similar comment as previous one.

Author Response

Response to Reviewer 2 Comments

1. Summary

2. Questions for General Evaluation

Reviewer’s Evaluation

Response and Revisions

Does the introduction provide sufficient background and include all relevant references?

Yes

Are all the cited references relevant to the research?

Yes

Is the research design appropriate?

Can be improved

Are the methods adequately described?

Must be improved

Are the results clearly presented?

Must be improved

Are the conclusions supported by the results?

Yes

3. Point-by-point response to Comments and Suggestions for Authors

Comments 1: Western blots are not quantified and only show a single lane for each group. These should be done in replicate and quantified to better support protein data. If the images are representative of the quants, then only showing single bands is fine.

Response 1: Thank you for pointing this out. We agree with this comment. All the protein expression detected by Western blots in our study have been repeated at least three times, and we chosed one of the results to show the expression of each protein, so the figures only showed a single lane.

Comments 2: At 36h PI, this MOI of CVB3 infection generally kills all the cells in culture. Thus it is difficult to ascertain if drops in protein levels are due to virally-mediated pathways so-to-speak, or if only cell corpses are being analyzed. Also, if YTHDF proteins are lower at this point, it would be interesting to see if this is transcriptional inhibition or if this may be due to proteolysis. It is also unclear at what point post-infection the other panels represent (and the rest of the figures). Additionally, single cell images are not much to go off of without any type of quantification assessing a larger population of cells.

4. Response to Comments on the Quality of English Language

Point 1: Fine.

Response 1:  Thanks.

5. Additional clarifications

No.

Reviewer 2 Report

Comments and Suggestions for Authors

The authors of this study have examined potential m6A modifications of the enterovirus coxsackievirus B3 (CVB3) RNA using a bioinformatics tool to determine that there were likely sites of methylation in the CVB3 genome including one site with highest likelihood.  In addition, the authors demonstrated that using 3-deazaadenosine (3-DAA) which decreases methylation by increasing the level of S-adenosylhomocysteine (SAH), a competitive inhibitor of S-adenosylmethionine (SAM).  Treatment of cells with this chemical resulted in a reduced level of CVB3 VP1, CVB3 plaque titer and GFP expression.  The authors state (lines 175-176) that these findings indicate that there are m6A modifications in CVB3 genomic RNA.  The fact that decreasing SAM decreases CVB3 replication is not proof of m6A modification of CVB3 genomic DNA.  At most it can be suggested that it is possible that CVB3 genomic RNA has sites which may be modified and that modification may alter the extent of viral replication. Given that CVB3 viral replication is affected by host factors which may be increased or decreased by SAM treatment, there are alternative explanations. It seems likely that alterations of the methylation on host mRNAs may alter the amount of essential host factors for enteroviruses resulting in alteration of the amount of virus replication in treated cells. 

To confirm the hypothesis of viral RNA methylation, the authors examined expression of writers, erasers and readers of m6A methylation and demonstrated that METTL3/14, FTO and ALKBH5 became cytoplasmic in CVB3 infected HeLa cells, a finding that might address the question of how a virus which replicates its genome in the cytoplasm could have genomic m6A modification.  In addition, the authors demonstrated that knockdowns of the METTL3/14 decreased viral replication, knockdowns of the methylases FTO and ALKBH5 increased viral replication and that knockdowns of two m6A binding proteins (YTH domain containing) decreased viral replication.  All of these indicate that the m6A methylation in host cells does affect viral replication but this does not specify m6A methylation of CVB3 RNA as, again, the methylation of host mRNAs is likely to affect the production of essential host factors for enterovirus replication.

To demonstrate the viral genomic modification, the authors mutated the DRACH site which was predicted to have the highest likelihood of m6A modification, nt4518, and demonstrated a significant difference in viral titer at 24 hours of replication in cell culture. It surprises me that a significant difference in viral titer occurs with C4519U as that variation is present in about 29% of the full length GenBank CVB3 sequences. It is true that the GenBank sequences are from viruses that do not have a GFP insert which may make virus replication more attenuated.  If m6A modification at this site has a significant effect upon replication, why is that C4519 not more conserved?  In addition, the authors need to state in the methods section that numbering in the genome is according to GenBank U57056.1, the entry associated with reference 24.  Can the authors address the question as to whether mutation of this site alters methylation of the RNA and, if that is significant, why this site is not conserved in natural isolates?

Overall, there is considerable suggestion in this study that the cell culture environment for CVB3 replication is affected by m6A methylation or at least by the pathways and enzymes that affect levels of methylation of RNA.  But there is not sufficient data provided by the authors to suggest that the genome of CVB3 has m6A methylation or that genomic methylation plays a part in CVB3 replication. 

Other issues:

The genome of CVB3 is approximately 7400 nt (without polyA), with the open reading frame extending from about nt742 to about nt7300.  Figure 1A shows the genome as extending to approximately 8100.  Please correct the X axis on this figure.

Please alter the legend to Figure 6: (C)Viral titres (PFU/ml) at 4,24,36,48 hpi. 

The legend to Figure 2B-H, should note that the 24-hour time point was used for these images.

Author Response

Response to Reviewer 2 Comments

1. Summary

2. Questions for General Evaluation

Reviewer’s Evaluation

Response and Revisions

Does the introduction provide sufficient background and include all relevant references?

Yes

Are all the cited references relevant to the research?

Yes

Is the research design appropriate?

Must be improved

Are the methods adequately described?

Can be improved

Are the results clearly presented?

Can be improved

Are the conclusions supported by the results?

Must be improved

3. Point-by-point response to Comments and Suggestions for Authors

Comments 1: The authors of this study have examined potential m6A modifications of the enterovirus coxsackievirus B3 (CVB3) RNA using a bioinformatics tool to determine that there were likely sites of methylation in the CVB3 genome including one site with highest likelihood.  In addition, the authors demonstrated that using 3-deazaadenosine (3-DAA) which decreases methylation by increasing the level of S-adenosylhomocysteine (SAH), a competitive inhibitor of S-adenosylmethionine (SAM).  Treatment of cells with this chemical resulted in a reduced level of CVB3 VP1, CVB3 plaque titer and GFP expression. The authors state (lines 175-176) that these findings indicate that there are m6A modifications in CVB3 genomic RNA.  The fact that decreasing SAM decreases CVB3 replication is not proof of m6A modification of CVB3 genomic DNA.  At most it can be suggested that it is possible that CVB3 genomic RNA has sites which may be modified and that modification may alter the extent of viral replication. Given that CVB3 viral replication is affected by host factors which may be increased or decreased by SAM treatment, there are alternative explanations. It seems likely that alterations of the methylation on host mRNAs may alter the amount of essential host factors for enteroviruses resulting in alteration of the amount of virus replication in treated cells.

To confirm the hypothesis of viral RNA methylation, the authors examined expression of writers, erasers and readers of m6A methylation and demonstrated that METTL3/14, FTO and ALKBH5 became cytoplasmic in CVB3 infected HeLa cells, a finding that might address the question of how a virus which replicates its genome in the cytoplasm could have genomic m6A modification.  In addition, the authors demonstrated that knockdowns of the METTL3/14 decreased viral replication, knockdowns of the methylases FTO and ALKBH5 increased viral replication and that knockdowns of two m6A binding proteins (YTH domain containing) decreased viral replication.  All of these indicate that the m6A methylation in host cells does affect viral replication but this does not specify m6A methylation of CVB3 RNA as, again, the methylation of host mRNAs is likely to affect the production of essential host factors for enterovirus replication.

To demonstrate the viral genomic modification, the authors mutated the DRACH site which was predicted to have the highest likelihood of m6A modification, nt4518, and demonstrated a significant difference in viral titer at 24 hours of replication in cell culture. It surprises me that a significant difference in viral titer occurs with C4519U as that variation is present in about 29% of the full length GenBank CVB3 sequences. It is true that the GenBank sequences are from viruses that do not have a GFP insert which may make virus replication more attenuated. If m6A modification at this site has a significant effect upon replication, why is that C4519 not more conserved?  In addition, the authors need to state in the methods section that numbering in the genome is according to GenBank U57056.1, the entry associated with reference 24.  Can the authors address the question as to whether mutation of this site alters methylation of the RNA and, if that is significant, why this site is not conserved in natural isolates?

Overall, there is considerable suggestion in this study that the cell culture environment for CVB3 replication is affected by m6A methylation or at least by the pathways and enzymes that affect levels of methylation of RNA. But there is not sufficient data provided by the authors to suggest that the genome of CVB3 has m6A methylation or that genomic methylation plays a part in CVB3 replication.

Response: Thank you very much for your comments. We carefully checked and modified our manuscript based on your kind suggestions

Response 1: Thank you for pointing this out. We have added the genome number is according to GenBank U57056.1 in Page 2, Line 84, the entry associated with reference 24. And as for the genome of CVB3 has m6A methylation, we may further use MeRIP-seq to detect the m6A methylation sites in CVB3 genome. C4519U mutation is without changing the amino acid but abolishing the m6a sites in CVB3 genome. However, the sites is different from that of wild type CVB3, because we predict m6A sites of the recombinant CVB3-GPF with a length about 8100 nt, not wild type CVB3 sequences which has genome about 7400 nt.  However, this m6a sites is chosen by SRAMP software prediction, and they may have so many other m6a sites in CVB3 genome, so we will use MeRIP-seq in future study to search the in-depth mechanism of how m6A methylation regulates CVB3 replication.

Comments 2: The genome of CVB3 is approximately 7400 nt (without polyA), with the open reading frame extending from about nt742 to about nt7300.  Figure 1A shows the genome as extending to approximately 8100. Please correct the X axis on this figure.

Response 2: Thank you for pointing this out. The Figure 1A is to show the genome length of the recombinant CVB3 which was inserted a GFP gene, so the total length is approximately 8100bp.

Comments 3: Please alter the legend to Figure 6: (C)Viral titres (PFU/ml) at 4,24,36,48 hpi.

Response 3: Thanks. We have altered the legend to Figure 6: (C)Viral titres (PFU/ml) at 4,24,36,48 hpi and marked in red in Page 9, Line 399.

Comments 4: The legend to Figure 2B-H, should note that the 24-hour time point was used for these images.

Response 4: Thank for your kind comments. We have added “at 24 h post infection(hpi)” in figure 2 legend and marked in red in Page 6, Line 240.

4. Response to Comments on the Quality of English Language

Point 1: Fine.

Response 1:  Thanks.

5. Additional clarifications

No.

Reviewer 3 Report

Comments and Suggestions for Authors

The manuscript “N6-Methyladenosine Positively Regulates Coxsackievirus B3 2 Replication” describes the study revealing the role of m6-adenosine and m6A methyltransferases as factors necessary for Coxsackie B3 virus replication.

The study has been carried up correctly, the methods used are selected adequately, and the conclusion corresponds to the results obtained.

One point in Introduction section is doubtful, namely the phrase in lines 62-63. “…ubiquitination of the 3D polymerase, which increases viral replication”. Despite these results are obtained by another researchers, this should be explained. Ubiquitination is a marker for further degradation of a protein by proteasomes. 3D polymerase is necessary for viral propagation. Why then degradation of polymerase results in increase of viral replication?

Section 2.4 (line 114). Please provide the temperature protocol for RT-PCR.

Section 2.5 (line 125). Please provide the sequence of shRNAs used for silencing of specific genes as well as nonspecific oligonucleotide.

After making clear these issues the manuscript can be published.

Author Response

Response to Reviewer 3 Comments

1. Summary

2. Questions for General Evaluation

Reviewer’s Evaluation

Response and Revisions

Does the introduction provide sufficient background and include all relevant references?

Yes

Are all the cited references relevant to the research?

Yes

Is the research design appropriate?

Yes

Are the methods adequately described?

Must be improved

Are the results clearly presented?

Yes

Are the conclusions supported by the results?

Yes

3. Point-by-point response to Comments and Suggestions for Authors

Comments 1: One point in Introduction section is doubtful, namely the phrase in lines 62-63. “…ubiquitination of the 3D polymerase, which increases viral replication”. Despite these results are obtained by another researchers, this should be explained. Ubiquitination is a marker for further degradation of a protein by proteasomes. 3D polymerase is necessary for viral propagation. Why then degradation of polymerase results in increase of viral replication?

Response 1: Thank you for pointing this out. we phrased the sentence to “During EV71 replication, METTL3 interacts with the viral RNA-dependent RNA polymerase 3D and induces increased SUMOylation and ubiquitination of the 3D that boosted viral replication, as previous study found that 3D is ubiquitinated in a SUMOylation-dependent manner that enhances the stability of the viral polymerase” in Page 2, Line 62-65. In a previous report (DOI: 10.1128/JVI.01756-16), they found that EV71 polymerase 3D was SUMOylated during EV71 infection, and mutation of SUMOylation sites impaired 3D polymerase activity and virus replication. Moreover, 3D is ubiquitinated in a SUMO-dependent manner, and SUMOylation is crucial for 3D stability, which may be due to the interaction of the two modifications thus to enhance virus replication.

Comments 2: Please provide the temperature protocol for RT-PCR.

Response 2: Thanks for your suggestions. We add it in the part 2.4. “Quantitative Reverse-Transcription PCR (qRT‒PCR)” as “Amplification was carried out at 95 °C for 3 min; 35 cycles of 95 °C for 15 s, 55 °C for 30 s, and 72 °C for 1 min; with a final extension of 72 °C for 10 min”in Page 3, Line 126-128.

Comments 3: Please provide the sequence of shRNAs used for silencing of specific genes as well as nonspecific oligonucleotide.

Response 3: Thanks. We added the sequence of shRNAs used for silencing of specific genes as well as nonspecific oligonucleotide in Part 2.5. in Page 3, Line 134-139.

4. Response to Comments on the Quality of English Language

Point 1: Fine.

Response 1:  Thanks.

5. Additional clarifications

No.

Round 2

Reviewer 2 Report

Comments and Suggestions for Authors

As the authors do not address actual m6A methylation of CVB3 RNA in this study, I still do not think they can say:
(Lines 186-187) “Taken together, these findings indicate that there are m6A modifications in CVB3 genomic RNA that can affect CVB3 replication.”

(Lines 454-456) “These results also indicate that m6A modification of nt4518 in the CVB3 genome affects the release of virus particles. Accordingly, m6A modification in the CVB3 genome plays an important role in CVB3 replication.”

The sole data for actual methylation is mutation of a single site (nt4519) in the viral genome which alters a motif for methylation and which results in a decrease of virus replication.  The authors state that this mutation does not alter viral protein.  As no virus sequence is given for the recombinant CVB3 used in this study and the authors state the numbering is different than in the GenBank U57056.1: (“However, the sites is different from that of wild type CVB3, because we predict m6A sites of the recombinant CVB3-GPF with a length about 8100 nt, not wild type CVB3 sequences which has genome about 7400 nt.”), it is very difficult to assess how this mutant virus might be altered in replication beyond the change in methylation.  At a minimum, the authors should provide the U57056.1 site which is equivalent to the nt4519 site or provide a GenBank accession for the actual virus sequence derived from pMKS1-eGFP-CVB3 or the sequence itself.

Nevertheless, they should alter the statement in lines 186 and 454 to state “these findings suggest” rather than “these findings indicate” and “These results also suggest” rather than “These results also indicate”.  This place in the discussion would be a good site to insert the statement that further work on methylation will permit the findings to be proven but the present findings demonstrate an effect of m6A methylation that is important for CVB3 replication. 

In addition, I failed to note in my earlier review that the authors stated in the first line of the abstract: “ Coxsackievirus B3 (CVB3) is the most common cause of viral myocarditis,” and in lines 34-35: “ Coxsackievirus B3 (CVB3) belongs to the family Picornaviridae and Enterovirus genus and has long been considered the most common cause of viral myocarditis”.  This has never been shown. While there is considerable evidence that enteroviruses are a common cause of human myocarditis based on detection in human myocarditic heart, coxsackievirus B3 is just one type of the Enterovirus B species. Present studies of myocarditic heart rarely identify the specific type but do often demonstrate the presence of enteroviruses.  Please restate this.  I suggest:

“Enteroviruses such as coxsackievirus B3 are identified as a common cause of viral myocarditis,”

“Coxsackievirus B3 (CVB3) belongs to the Enterovirus genus of the Picornaviridae and has been long studied in models of viral myocarditis”. 

Author Response

Response to Reviewer 2 Comments

1. Summary

2. Questions for General Evaluation

Reviewer’s Evaluation

Response and Revisions

Does the introduction provide sufficient background and include all relevant references?

Yes

Are all the cited references relevant to the research?

Yes

Is the research design appropriate?

Yes

Are the methods adequately described?

Must be improve

Are the results clearly presented?

Can be improved

Are the conclusions supported by the results?

Must be improved

3. Point-by-point response to Comments and Suggestions for Authors

Comments 1: As the authors do not address actual m6A methylation of CVB3 RNA in this study, I still do not think they can say:

(Lines 186-187) “Taken together, these findings indicate that there are m6A modifications in CVB3 genomic RNA that can affect CVB3 replication.”

Response 1: Thank you for pointing this out. We agree with this comment. As the m6a sites were predicted by software, not by MeRIP-seq, so it dose not address actual m6A methylation of CVB3 RNA in this study; However,we found 3-deazaadenosine (3-DAA), a m(6)A modification inhibitor could inhibit CVB3 replication. Therefore, we changed this sentence to" these findings indicate m6A modifications may affect CVB3 replication".

Comments 2: (Lines 454-456) “These results also indicate that m6A modification of nt4518 in the CVB3 genome affects the release of virus particles. Accordingly, m6A modification in the CVB3 genome plays an important role in CVB3 replication.”

The sole data for actual methylation is mutation of a single site (nt4519) in the viral genome which alters a motif for methylation and which results in a decrease of virus replication.  The authors state that this mutation does not alter viral protein. As no virus sequence is given for the recombinant CVB3 used in this study and the authors state the numbering is different than in the GenBank U57056.1: (“However, the sites is different from that of wild type CVB3, because we predict m6A sites of the recombinant CVB3-GPF with a length about 8100 nt, not wild type CVB3 sequences which has genome about 7400 nt.”), it is very difficult to assess how this mutant virus might be altered in replication beyond the change in methylation.  At a minimum, the authors should provide the U57056.1 site which is equivalent to the nt4519 site or provide a GenBank accession for the actual virus sequence derived from pMKS1-eGFP-CVB3 or the sequence itself.

Nevertheless, they should alter the statement in lines 186 and 454 to state “these findings suggest” rather than “these findings indicate” and “These results also suggest” rather than “These results also indicate”.  This place in the discussion would be a good site to insert the statement that further work on methylation will permit the findings to be proven but the present findings demonstrate an effect of m6A methylation that is important for CVB3 replication.

Response 2: Thanks for your kind suggestions. We have changed the sentence lines 186 and 454 accordingly.

Comments 3: In addition, I failed to note in my earlier review that the authors stated in the first line of the abstract: “Coxsackievirus B3 (CVB3) is the most common cause of viral myocarditis,” and in lines 34-35: “Coxsackievirus B3 (CVB3) belongs to the family Picornaviridae and Enterovirus genus and has long been considered the most common cause of viral myocarditis”.  This has never been shown. While there is considerable evidence that enteroviruses are a common cause of human myocarditis based on detection in human myocarditic heart, coxsackievirus B3 is just one type of the Enterovirus B species. Present studies of myocarditic heart rarely identify the specific type but do often demonstrate the presence of enteroviruses.  Please restate this.  I suggest:

“Enteroviruses such as coxsackievirus B3 are identified as a common cause of viral myocarditis,”

“Coxsackievirus B3 (CVB3) belongs to the Enterovirus genus of the Picornaviridae and has been long studied in models of viral myocarditis”.

Response 3: Thanks. We have changed the sentence to in the first line of abstract to: "Enteroviruses such as coxsackievirus B3 are identified as a common cause of viral myocarditis" and "Coxsackievirus B3 (CVB3) belongs to the Enterovirus genus of the Picornaviridae and has been long studied in models of viral myocarditis" in line 34-35.

4. Response to Comments on the Quality of English Language

Point 1: Fine

Response 1:  Thanks.

5. Additional clarifications

No.